# Identification of a New HIV-1 BC Intersubtype Circulating Recombinant Form (CRF108_BC) in Spain

**DOI:** 10.3390/v13010093

**Published:** 2021-01-12

**Authors:** Javier E. Cañada, Elena Delgado, Horacio Gil, Mónica Sánchez, Sonia Benito, Elena García-Bodas, Carmen Gómez-González, Andrés Canut-Blasco, Joseba Portu-Zapirain, Ester Sáez de Adana, Mireia De la Peña, Sofía Ibarra, Gustavo Cilla, José Antonio Iribarren, Ana Martínez-Sapiña, Michael M. Thomson

**Affiliations:** 1HIV Biology and Variability Unit, Centro Nacional de Microbiología, Instituto de Salud Carlos III, Majadahonda, 28220 Madrid, Spain; jecanada@isciii.es (J.E.C.); delgade@isciii.es (E.D.); hgil@isciii.es (H.G.); monicasanchez@isciii.es (M.S.); sbenito@isciii.es (S.B.); egbodas@isciii.es (E.G.-B.); 2Department of Microbiology, Hospital Universitario Araba, 01009 Vitoria-Gasteiz, Spain; carmen.gomezgonzalez@osakidetza.eus (C.G.-G.); andres.canutblasco@osakidetza.eus (A.C.-B.); 3Bioaraba, Infectious Diseases Research Group, 01009 Vitoria-Gasteiz, Spain; josejoaquin.portuzapirain@osakidetza.eus; 4Department of Infectious Diseases-Internal Medicine, Hospital Universitario Araba, 01009 Vitoria-Gasteiz, Spain; ester.saezdeadanaarroniz@osakidetza.eus; 5Department of Infectious Diseases, Hospital Universitario Basurto, 48013 Bilbao, Spain; mireia.delapenatrigueros@osakidetza.eus (M.D.l.P.); sofia.ibarraugarte@osakidetza.eus (S.I.); 6Biodonostia, Department of Microbiology, Hospital Universitario Donostia, 20080 San Sebastián, Spain; carlosgustavosantiago.cillaeguiluz@osakidetza.eus; 7Biodonostia, Department of Infectious Diseases, Hospital Universitario Donostia, 20080 San Sebastián, Spain; joseantonio.iribarrenloyarte@osakidetza.eus; 8Department of Microbiology, Hospital Universitario Miguel Servet, 50009 Zaragoza, Spain; amartinezsa@salud.aragon.es

**Keywords:** HIV-1, circulating recombinant form, HIV-1 genetic diversity, HIV-1 phylogeny, HIV-1 molecular epidemiology

## Abstract

The extraordinary genetic variability of human immunodeficiency virus type 1 (HIV-1) group M has led to the identification of 10 subtypes, 102 circulating recombinant forms (CRFs) and numerous unique recombinant forms. Among CRFs, 11 derived from subtypes B and C have been identified in China, Brazil, and Italy. Here we identify a new HIV-1 CRF_BC in Northern Spain. Originally, a phylogenetic cluster of 15 viruses of subtype C in protease-reverse transcriptase was identified in an HIV-1 molecular surveillance study in Spain, most of them from individuals from the Basque Country and heterosexually transmitted. Analyses of near full-length genome sequences from six viruses from three cities revealed that they were BC recombinant with coincident mosaic structures different from known CRFs. This allowed the definition of a new HIV-1 CRF designated CRF108_BC, whose genome is predominantly of subtype C, with four short subtype B fragments. Phylogenetic analyses with database sequences supported a Brazilian ancestry of the parental subtype C strain. Coalescent Bayesian analyses estimated the most recent common ancestor of CRF108_BC in the city of Vitoria, Basque Country, around 2000. CRF108_BC is the first CRF_BC identified in Spain and the second in Europe, after CRF60_BC, both phylogenetically related to Brazilian subtype C strains.

## 1. Introduction

HIV-1 is characterized by high mutation and recombination rates, which have led to the generation of extraordinary genetic diversity. Four HIV-1 groups have been characterized: M, N, O, and P. Group M, the oldest lineage [1,2], is the causative of the global pandemic and is subdivided into ten subtypes (A–D, F–H, J–L), of which subtype C is the most prevalent worldwide, circulating mainly in Southern and East Africa, India, and Southern Brazil, and subtype B are the most prevalent in Western Europe and the Americas [3].

Recombination between subtypes has led to the generation of circulating and unique recombinant forms (CRFs and URFs, respectively), which represent around 23% of HIV-1 strains worldwide [3]. To define a CRF, at least three HIV-1 near full-length genomes (NFLGs) must be characterized from epidemiologically unlinked individuals, showing identical mosaic patterns and clustering in phylogenetic trees apart from previously defined CRFs [4]. To date, a total of 102 CRFs have been reported in the literature.

HIV-1 was introduced in Western Europe in the early 1980s among men who have sex with men (MSM) and persons who inject drugs (PWID) infected with viruses of subtype B, which is the current predominant genetic form (67.2%), followed by subtype C (5.3%) [5]. Among non-subtype B clades, there are several CRFs first identified in Western Europe: CRF04_cpx [6], CRF14_BG [7,8], CRF42_BF [9], CRF47_BF [10], CRF50_A1D [11], CRF56_cpx [12], CRF60_BC [13,14], CRF73_BG [15], CRF94_cpx [16] and CRF98_06B [17].

In this study, we report the first CRF_BC identified in Spain and the second in Europe, estimating its most probable origin by phylogeographic and phylodynamic analyses.

## 2. Materials and Methods

### 2.1. Patients

Plasma or whole blood samples were collected in 1999–2020 from more than 13,000 HIV-1-infected individuals from 10 Spanish regions for determination of antiretroviral drug resistance mutations and for molecular epidemiological surveillance of HIV-1.

### 2.2. Nucleic Acid Extraction, Amplification and Sequencing

RNA was extracted from 1 mL plasma using NucliSENS^®^ EasyMAG^®^ (bioMérieux, Marcy l’Etoile, France). DNA was extracted from 200 µL whole blood using QIAamp^®^ DNA DSP blood mini kit (Qiagen, Hilden, Germany), following the manufacturer’s instructions. A protease-reverse transcriptase (PR–RT) fragment of pol (HXB2 positions 2253–3629) was amplified by RT–PCR followed by nested-PCR from RNA or by nested-PCR from DNA, as previously described [18].

NFLGs were amplified from plasma RNA, with PR–RT previously sequenced, through RT–PCR followed by nested-PCR in five overlapping fragments, as reported previously in [7,19] and modified from [20]. The amplification strategy is schematically depicted in Appendix A, and PCR primers are listed in Appendix A. Sequencing was done with an automated capillary sequencer. Fifteen PR–RT, six NFLG, and one semigenome sequence were deposited in GenBank (Table 1).

### 2.3. Phylogenetic Analyses

Initial phylogenetic analyses were performed with FastTree v2.1 [21] with more than 16,000 HIV-1 PR–RT sequences obtained in our laboratory from more than 13,000 individuals whose samples were collected in Spain in 1999–2020, similar sequences retrieved through BLAST searches [22] from the Los Alamos HIV Sequence Database [23], and subtype and CRF references. For these analyses, the general time-reversible with CAT approximation for rate heterogeneity among sites (GTR + CAT) substitution model was used, with the assessment of node support with Shimodaira–Hasegawa (SH)-like local support values.

Subsequent maximum-likelihood (ML) analyses were performed with W-IQ-Tree [24], including sequences similar to the identified cluster, retrieved from the HIV Sequence Database [23] through BLAST searches [22]. These analyses were performed with a 1200 nt fragment of the PR–RT region (HXB2 positions 2253–3452). The substitution model was GTR with gamma-distributed heterogeneity across sites, allowing for a proportion of invariant sites (GTR + G + I), and node support was assessed through ultrafast bootstrapping with 1000 replicates.

The recombination patterns in NFLGs were determined by bootscanning [25] using Simplot v.3.5.1 [26], including HIV-1 subtype references downloaded from the Los Alamos HIV Sequence Database, with a 250 nt window moving in 20 nt steps and trees constructed with the neighbor-joining method and Kimura 2-parameter substitution model. Breakpoints were located more precisely through sequence inspection by determining the segment where similarity of the BC recombinants with NFLG genomes of subtype B and Brazilian subtype C viruses changed between clades. Breakpoints were located at the midpoint between two adjacent subtype-discriminating nucleotides (defined as those differing between subtype consensuses and present in >75% viruses of one of the parental clades and in <10% of the other) where similarity changed between subtypes.

### 2.4. Phylogeographic and Phylodynamic Analyses

The time and location of the most recent common ancestor (MRCA) were estimated with the Bayesian coalescent Markov Chain Monte Carlo (MCMC) method, implemented in BEAST v1.10.4 [27], summarizing the set of trees of the posterior distribution in a maximum clade credibility (MCC) tree. PR–RT sequences (1.2 kb) of the identified cluster were used, labeled with the location and the year of collection of the sample. Fifty subtype C sequences from different countries and collection years were included to provide a temporal signal, previously assessed with TempEst v1.5.1 [28]. We chose an HKY substitution model with gamma-distributed among-site rate heterogeneity and two partitions in codon positions (1st + 2nd; 3rd) [29]. A Bayesian skyline coalescent model was chosen with a lognormal uncorrelated relaxed clock model. Uniform priors were used for absolute substitution rates (0–0.02 sub/site/year). MCMC analyses were run for 70 million generations, sampling every 4000 generations. Tracer v1.7.1 [30] was used to check MCMC convergence, ensuring effective sample sizes (ESS) of all parameters above 200 [30]. Trees were visualized with FigTree v.1.3.1 (Rambaut, http://tree.bio.ed.aC.uk/software/figtree/).

### 2.5. Antiretroviral Drug Resistance Analysis

Antiretroviral drug resistance was analyzed with the HIVdb program at Stanford University’s HIV Drug Resistance Database [31].

## 3. Results

### 3.1. PR–RT Sequence Analyses

The phylogenetic analyses of HIV-1 PR–RT sequences from our cohort identified a monophyletic cluster of 15 viruses of subtype C supported by an SH-like value of 1, which was designated C_2. The ML tree constructed with W-IQ-Tree, including sequences retrieved from databases through BLAST similarity searches, revealed no additional viruses branching within the cluster and its relationship to viruses of the subtype C strain circulating in Brazil (Figure 1). Epidemiological and clinical data of the 15 patients of the C_2 cluster are summarized in Table 1. Most of them were Spanish, except a Brazilian individual, and were diagnosed with HIV-1 infection from 2006 to 2019 in the Basque Country (10 in the city of Vitoria), except one patient diagnosed in Zaragoza. Ten patients were men, and 5 were women; heterosexual transmission was reported in 12 (80%) infections, and one patient was a self-reported MSM.

No antiretroviral drug resistance mutations were found in any of the PR–RT sequences.

### 3.2. NFLG Sequence Analyses

To determine whether the viruses grouping in C_2 were of uniform subtype along their genomes or recombinant, six NFLG sequences and a semigenome obtained from viruses collected in three cities were analyzed by bootscanning. The analyses showed that the viruses were BC recombinant, with eight breakpoints delimiting four short subtype B fragments, located in *pol*, *pol-vif* overlap, *vif-vpr-tat* overlap, and *nef*, respectively, in a genome predominantly of subtype C (Figure 2A). The mosaic structure inferred from the bootscan analyses complemented with sequence inspection to define more precisely breakpoint locations is shown in Figure 2B. In an ML phylogenetic tree, all six newly derived NFLG of the BC recombinant viruses are grouped in a cluster separate from previously identified CRF_BCs (Figure 3). These results allow defining a new HIV-1 CRF, which was designated CRF108_BC. Comparison of CRF108_BC’s mosaic structure with those of previously identified CRF_BCs is shown in Appendix A.

### 3.3. Phylogeographic and Phylodynamic Analyses

To estimate the temporal and geographic origin of CRF108_BC, a Bayesian coalescent analysis was performed with the 15 PR-RT sequences of the C_2 cluster and 50 subtype C database sequences from different countries for which year and location of sample collection were available. Prior to this analysis, the existence of a temporal signal was checked with TempEst v1.5.1, which revealed a clock-like structure in the data set (r^2^ = 0.38), indicating sufficient temporal signal to perform the analyses.

The tMRCA of the cluster was estimated around 2000 (95% HPD, 1995–2004) in the city of Vitoria with a location posterior probability of 0.998. An ancestry in Brazil was also strongly supported (Figure 5).

## 4. Discussion

The characterization of six HIV-1 NFLG sequences of BC recombinant viruses obtained from epidemiologically-unlinked patients showing a coincident mosaic structure different from previously identified CRFs and clustering with a 100% bootstrap value allowed to define a new CRF, designated CRF108_BC. Nine additional partial sequences are grouped in a monophyletic cluster in PR-RT. All 15 patients were diagnosed with HIV-1 infection in Northern Spain between 2006 and 2019 and were infected predominantly via heterosexual contact.

CRF108_BC is the first CRF recombinant of subtypes B and C parental strains identified in Spain and the second in Europe (after CRF60_BC in Italy [13]). Ten CRF_BCs have been identified elsewhere, nine in China [CRF07_BC [32], CRF08_BC [33], CRF57_BC [34], CRF61_BC [35], CRF62_BC [36], CRF64_BC [37], CRF85_BC [38], CRF86_BC [39] and CRF88_BC [40]] and one in Brazil [CRF31_BC [41]]. Similar to CRF60_BC, the subtype C parental strain of CRF108_BC is phylogenetically related to the subtype C strain circulating in Brazil. However, the recombination event giving rise to CRF108_BC could have occurred either in Brazil, where both B and C subtypes co-circulate at high proportions in some areas [42,43] or in Spain, since we could not track the ancestry of the parental subtype B strain to any country. A Spanish origin of CRF108_BC would be supported by an inferred Spanish tMRCA in the city of Vitoria, Basque Country, Northern Spain, around 2000. However, we cannot rule out that CRF108_BC could be circulating at a low prevalence in some area(s) of Brazil, which could explain its lack of representation in public sequence databases.

CRF108_BC, similarly to all other CRF_BCs identified to date, is predominantly of subtype C. It is interesting to note that in all CRF_BCs, *env* is mostly of subtype C, which could lead to the speculation that a subtype C envelope could confer evolutionary advantages regarding viral fitness, escape to immune response or a more efficient replication or transmission over a subtype B envelope. Further investigations are required to confirm any of these possibilities.

To date, the expansion of CRF108_BC has taken place from 2006 to 2019, with 53% of cases diagnosed in the last 4 years, and limited to a small geographical area in Spain, with 10 of 15 cases in the city of Vitoria and 4 other cases in two neighboring provinces of the Basque Country. Outside of the Basque Country, we only have found a single infection with CRF108_BC in the city of Zaragoza among HIV-1 sequences from more than 13,000 patients from 10 regions of Spain analyzed by us and all sequences from public databases.

In Asia, the majority of CRF_BCs were transmitted initially among persons who inject drugs (PWID) [40,44] and recently expanded among MSM [45]. In Europe, CRF60_BC is associated with propagation among MSM [13]. Although, at present, MSM is the most frequent transmission route of HIV-1 in Spain [46], two CRFs previously identified in Spain, CRF14_BG and CRF73_BG, propagated mainly among PWID [7,15]. A third CRF identified in Spain, CRF47_BF, was associated with heterosexual transmission [10], similarly to CRF108_BC. However, we have observed further propagation of CRF47_BF among MSM [47]. This pattern could potentially be repeated with CRF108_BC since, along with heterosexually transmitted cases (with 27% women), we find an MSM diagnosed in 2017. This suggests that transmission networks of HIV-1 in Spain among heterosexuals may be drifting towards MSM. We have also observed the reverse situation in the case of a CRF02_AG cluster spreading from MSM to a heterosexual network [48].

NFLG sequencing of HIV-1 strains involved in expanding transmission clusters is highly recommendable in order to identify new CRFs, which may have acquired adaptive advantages through recombination [49,50] even when recombination is not suspected in partial sequences, as it occurs with CRF108_BC. Genetic diversity and recombination are major obstacles to the development of an effective vaccine against HIV-1 [51,52,53]. Characterization and molecular epidemiological surveillance of expanding new HIV-1 CRFs may play an important role in public health actions, including the selection of optimal immunogens for effective vaccines.

## 5. Conclusions

A new HIV-1 CRF derived from subtypes B and C, designated CRF108_BC, has been identified after the analysis of six NFLG from three cities in Northern Spain, which was originally identified as a subtype C cluster in PR–RT sequences comprising 15 individuals. Phylogenetic and phylogeographic analyses point to a Brazilian ancestry, although it is unclear whether the recombination event took place in Brazil or in Spain.

Among CRFs derived from B and C subtypes, CRF108_BC is the 12th identified, the first in Spain and the second in Europe. Its spread is currently limited, with only 15 cases detected, most of them transmitted via heterosexual contact. Considering that more than half of them were diagnosed in the last four years, molecular epidemiological surveillance seems justified to examine further spread. The results of this study also advocate for NFLG sequence characterization of emerging HIV-1 clusters, which may represent new CRFs.

## Figures and Tables

**Figure 1 viruses-13-00093-f001:**
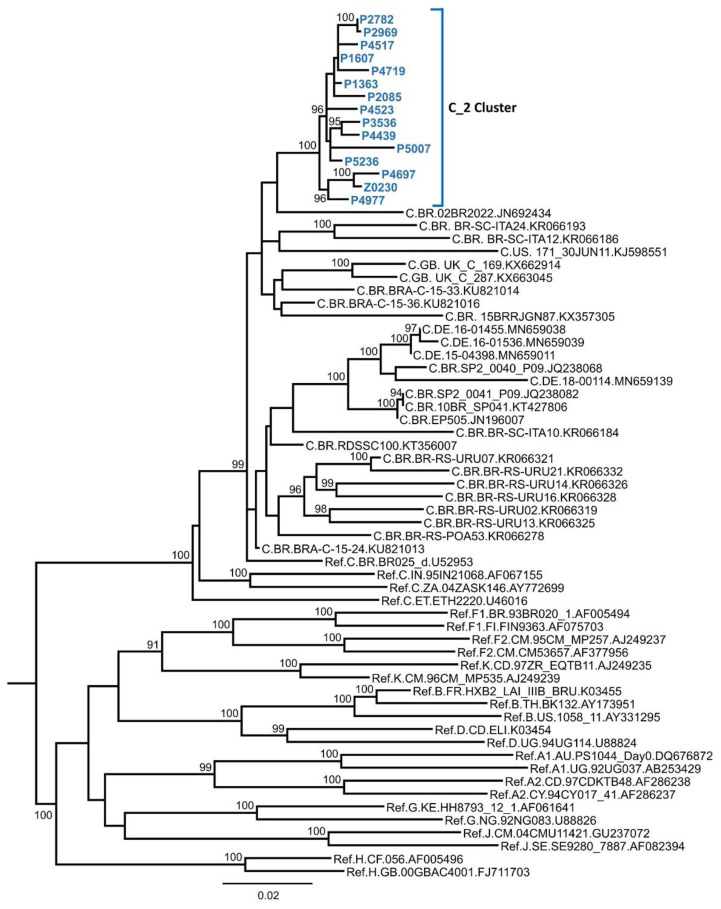
Maximum likelihood tree of C_2 cluster. Similar sequences retrieved from databases and subtype references are also included. Only bootstrap values ≥90% are shown. Sequences obtained in our laboratory are in blue and bold type. Database sequences are labeled with subtype, two-letter ISO code of the country of collection, virus names, and GenBank accession. Subtype reference sequences are labeled with “Ref.”.

**Figure 2 viruses-13-00093-f002:**
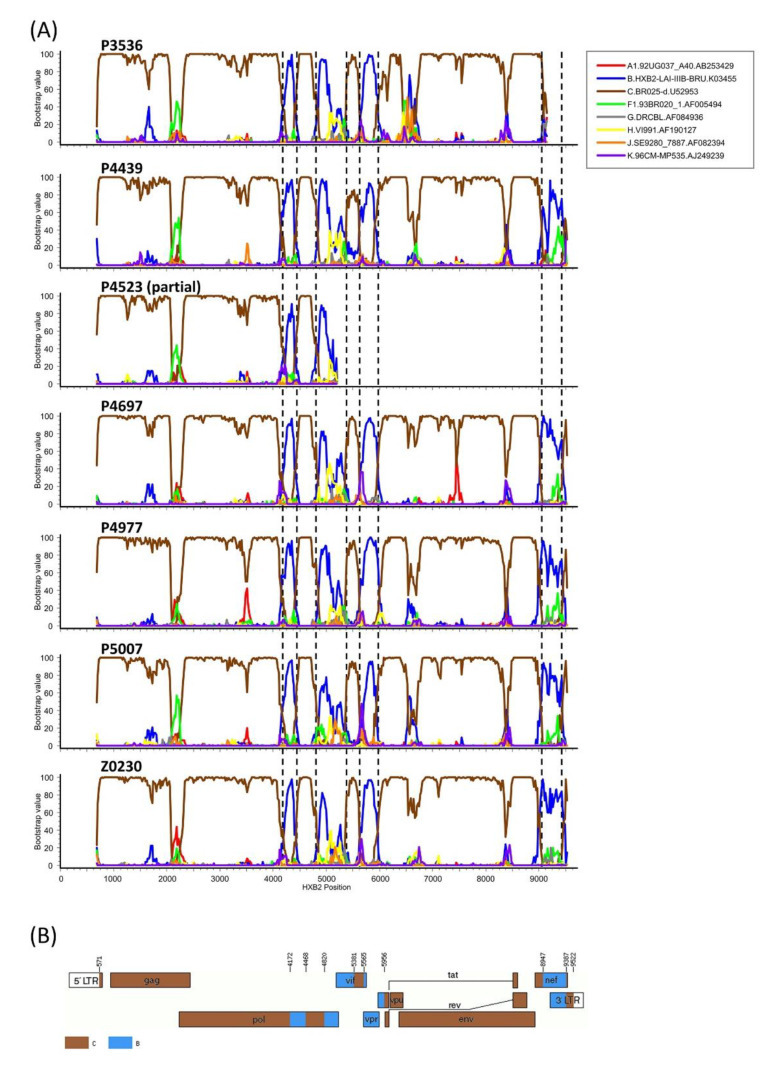
(**A**) Bootscan analyses of 6 NFLG and 1 semigenome sequences of viruses of the C_2 cluster. The horizontal axis represents the position in the HXB2 genome of the midpoint of a 250 nt window moving in 20 in increments, and the vertical axis represents the bootstrap value supporting clustering of the query sequence with subtype references. Vertical dashed lines denote breakpoint locations; (**B**) Mosaic structure of HIV-1 BC intersubtype circulating recombinant form (CRF108_BC). Breakpoint positions in the HXB2 genome are indicated.

**Figure 3 viruses-13-00093-f003:**
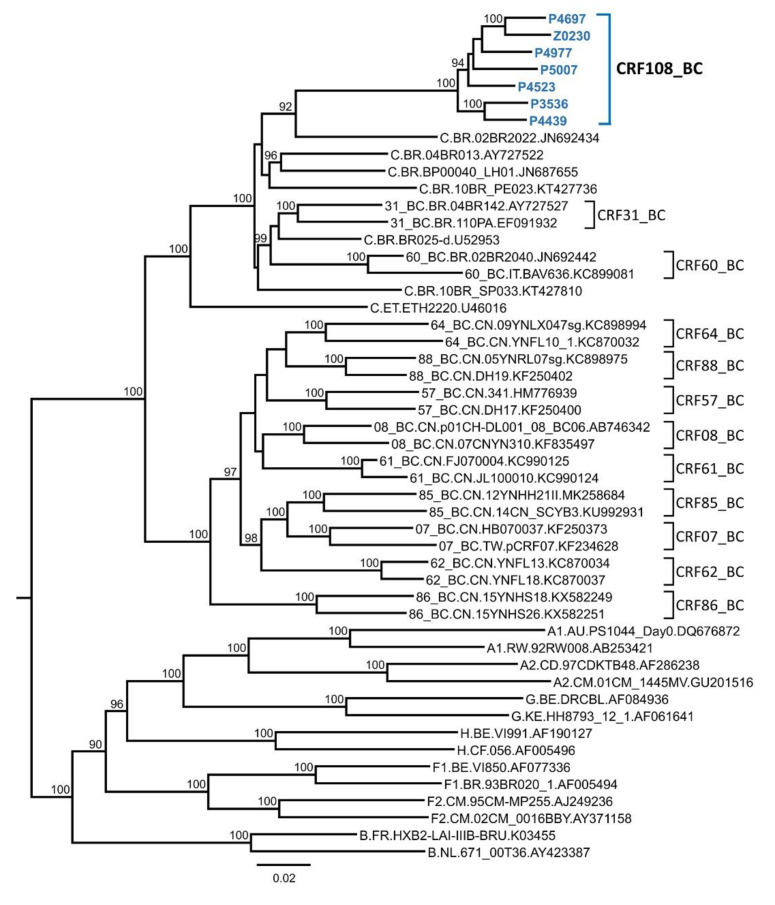
Maximum likelihood tree of NFLGs of CRF108_BC and all CRF_BCs identified to date. Only bootstrap values ≥90% are shown. CRF108_BC sequences are in blue and bold type. Database Scheme 108. BC, BLAST searches for similar sequences were done with subtype C and B fragments at the Los Alamos HIV Sequence Database. With regard to the subtype C fragments, searches were done with the two largest fragments in *gag-pol* and *tat-rev-vpu-env-nef*, respectively. A phylogenetic tree with all subtype C concatenated fragments from the NFLGs of CRF108_BC, and most similar database sequences showed the closest relationship with the Brazilian virus 02BR2022 from Sao Paulo (Figure 4). Similarity searches with the four subtype B fragments and subsequent phylogenetic analyses with individual or concatenated fragments failed to identify any database virus related to the subtype B parental strain of CRF108_BC.

**Figure 4 viruses-13-00093-f004:**
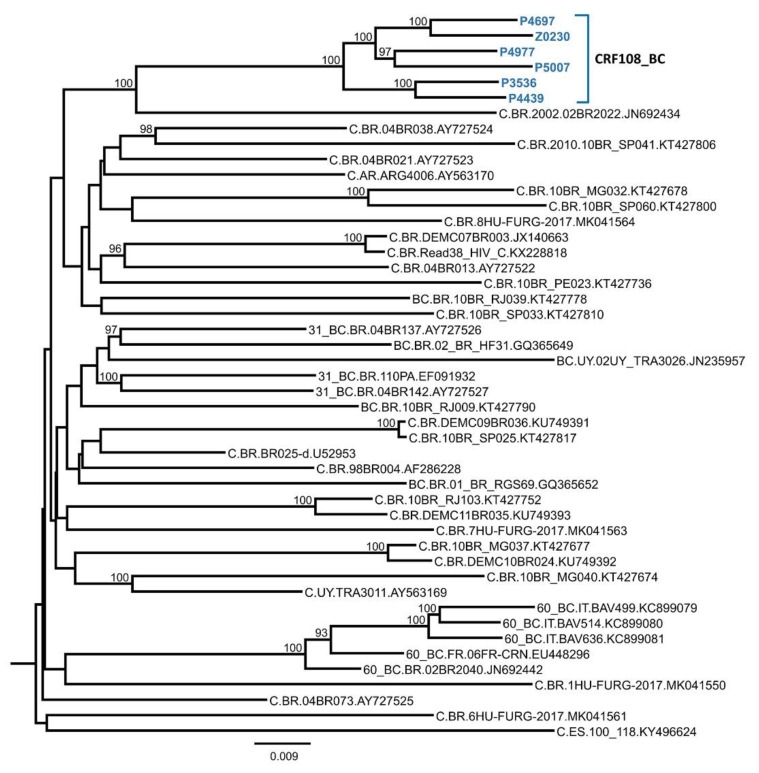
Phylogenetic tree of concatenated subtype C fragments of CRF108_BC. Only bootstrap values ≥90% are shown. CRF108_BC sequences are in blue and bold type. Database sequences are labeled with subtype, two-letter ISO code of the country of sample collection, virus name, and GenBank accession.

**Figure 5 viruses-13-00093-f005:**
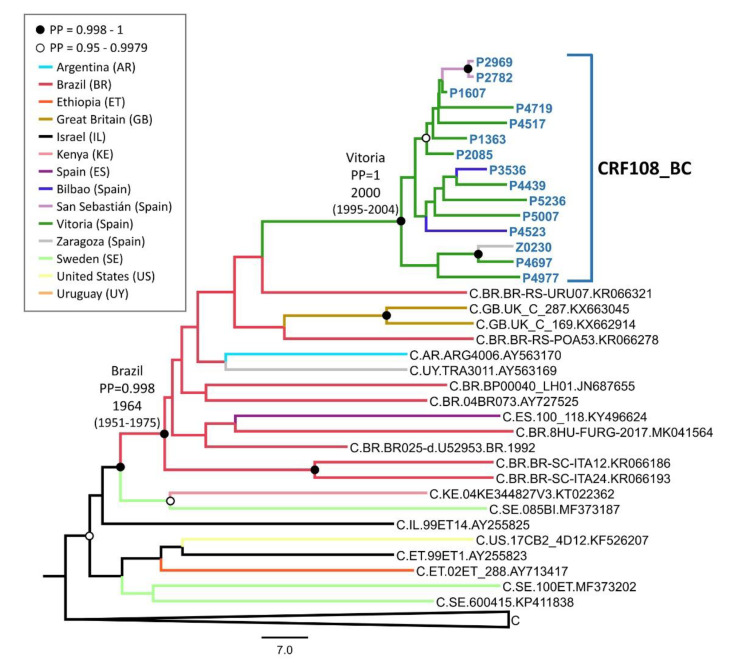
Maximum clade credibility tree of PR–RT sequences of CRF108_BC and 50 subtype C sequences from databases. Sequences belonging to the CRF108_BC cluster are in blue and bold type. Database sequences are labeled with subtype, two-letter ISO country-code, virus name, and GenBank accession. Colors of terminal and internal branches represent sampling locations and most probable locations of the corresponding nodes, respectively, according to the legend. For the nodes corresponding to the cluster and its closest ancestor, the most probable locations and the mean tMRCA (with 95% HPD intervals) are indicated. Nodes supported by PP (posterior probability) = 0.998–1 and PP = 0.95–0.9979 are marked with filled and unfilled circles, respectively. 29 branches corresponding to sequences from Botswana, Cyprus, Ethiopia, Georgia, Israel, Kenya, Malawi, Senegal, South Africa, Sweden, Tanzania, United Kingdom, Yemen, and Zambia were collapsed for better viewing.

**Table 1 viruses-13-00093-t001:** Epidemiological and clinical data of the patients and GenBank accessions of sequences.

Sample	City	Region *	Country of Origin	Year of Diagnosis	Year of Collection	Gender	Age	Transmission Route †	PR–RT GenBank Accession	NFLG GenBank Accession
P1363	Vitoria	Basque C.	Spain	2006	2010	M	64	Het	MT436238	-
P1607	Vitoria	Basque C.	Brazil	2007	2007	F	20	Trans	MT436239	-
P2085	Vitoria	Basque C.	Spain	2008	2008	M	40	Het	MT559129	-
P2782	San Sebastián	Basque C.	Spain	2011	2011	M	38	Het	MT436240	-
P2969	San Sebastián	Basque C.	Spain	2011	2011	F	32	Het	MT436241	-
P3536	Bilbao	Basque C.	Spain	2013	2013	M	70	Het	MT436242	MT559130
P4439	Vitoria	Basque C.	Spain	2016	2016	F	58	Het	MT436244	MN172222
P4517	Vitoria	Basque C.	Spain	2010	2016	M	58	n.a.	MT436245	-
P4523	Bilbao	Basque C.	Spain	2016	2016	M	42	Het	MT436246	MT559131 ‡
P4697	Vitoria	Basque C.	Spain	2017	2017	F	28	Het	MT436247	MT559132
P4719	Vitoria	Basque C.	Spain	2017	2017	M	44	Het	MT436248	-
P4977	Vitoria	Basque C.	Spain	2018	2018	M	53	Het	MT436249	MN172223
P5007	Vitoria	Basque C.	Spain	2018	2018	M	35	Het	MT436250	MN172224
P5236	Vitoria	Basque C.	Spain	2019	2019	F	65	Het	MT436251	-
Z0230	Zaragoza	Aragon	Spain	2017	2017	M	36	MSM	MT436252	MN172225

* Basque.C.: Basque Country. † Het: heterosexual; MSM: men who have sex with men; Trans: transsexual; n.a.: not available. ‡ Semigenome.

## Data Availability

The sequences newly obtained for this study are openly available at GenBank (https://www.ncbi.nlm.nih.gov/genbank/) under accessions MT436238-MT436252, MT559130-MT559132, MN172222-MN172225.

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
