# Peer review of "Identification of a New HIV-1 BC Intersubtype Circulating Recombinant Form (CRF108_BC) in Spain"

_viruses, 2021, doi:10.3390/v13010093_

Round 1
Reviewer 1 Report
The paper is well written and data and figures are clearly presented.
They identified a new intersubtype HIV-1 CRF, called CRF108_BC, whose
genome is predominantly of subtype C but has 4 short subtype B fragments.
CRF108_BC is the first CRF_BC that is identified in Spain and the second
in Europe, making this paper relevant for further analysis of patient
samples and spread of novel isolates and CRFs of HIV-1.
Also considering that more than half of them were diagnosed in the last
four years, molecular epidemiological surveillance seems also justified,
as they concluded, to examine further spread of these viruses.
Reviewer 2 Report
This work describes the discovery and phylogenetic characterization of a new circulating recombinant form of HIV-1 in northern Spain, and between subtypes C and B, dubbed CRF108_BC. The authors use rigorous analytical methods to identify breakpoints between the predominantly type C backbone and islands of subtype B sequence, and also show that the discovered form is distinct from BC recombinants previously identified. Moreover, they propose and are able to reasonably defend the hypothesis that the recombination event giving rise to this form may have occured in Brazil, circa 2000.
The work is very well written and the data clearly presented, and the primary finding is of significant interest to specialists in HIV-1 phylogeny. As a matter of specific interest, one wonders why the the new form was named CRF108_BC, if only 102 CRFs had been described previously.